# Regularizing Black-box Models for Improved Interpretability

## Abstract

Most of the work on interpretable machine learning has focused on designing either inherently interpretable models, which typically trade-off accuracy for interpretability, or post-hoc explanation systems, which lack guarantees about their explanation quality. We explore an alternative to these approaches by directly regularizing a black-box model for interpretability at training time. Our approach explicitly connects three key aspects of interpretable machine learning: (i) the model's inherent interpretability, (ii) the explanation system used at test time, and (iii) the metrics that measure explanation quality. Our regularization results in substantial improvement in terms of the explanation fidelity and stability metrics across a range of datasets and black-box explanation systems while slightly improving accuracy. Finally, we justify theoretically that the benefits of our regularization generalize to unseen points.

## 1 Introduction

Complex learning-based systems are increasingly shaping our daily lives, and, in order to monitor and understand these systems, we require clear explanations of model behavior. While model interpretability has many definitions and is often largely application specific (Lipton, 2016), local explanations are a popular and powerful tool (Ribeiro et al., 2016). Recent work on local interpretability in machine learning ranges from proposals of new models that are interpretable *by-design* (*e.g.*, Wang and Rudin, 2015; Caruana et al., 2015) to model-agnostic *post-hoc* algorithms for interpreting complex, black-box predictors such as ensembles and deep neural networks (*e.g.*, Ribeiro et al., 2016; Lei et al., 2016; Lundberg and Lee, 2017; Selvaraju et al., 2017; Kim et al., 2018). Despite the variety of technical approaches, the underlying goal of all of these works is to develop an interpretable predictive system that produces two outputs: a prediction and its underlying explanation.

Both interpretability by-design and post-hoc explanation strategies have limitations. On one hand, the by-design approaches are restricted to working with model families that provide inherent interpretability, potentially at the cost of accuracy. On the other hand, by performing two disjoint steps, there is no guarantee that post-hoc explainers applied to an arbitrary model will produce explanations of suitable quality. Moreover, recent approaches that claim to overcome this apparent trade-off between prediction accuracy and explanation quality are in fact by-design proposals that impose certain constraints on the underlying model families they consider (*e.g.*, Al-Shedivat et al., 2017; Plumb et al., 2018; Alvarez-Melis and Jaakkola, 2018a). In this work, we propose a novel alternative strategy called *Explanation-based Optimization* (ExpO) that aims to address both of these shortcomings by adding an *interpretability regularizer* to the loss function of an arbitrary predictive model. We illustrate how ExpO can influence the interpretability and accuracy of a model in Figure 1 (left).

**Illustration.** Consider a situation where Bob's loan application is denied by a machine learning system; see the toy illustration in Figure 1 (right). Here, a good local explanation can help Bob understand how to improve his application in order to get the loan. Unfortunately, a standard model, a multi-layer perceptron trained with SGD, is not very interpretable because it has many abrupt changes. Indeed, we can quantitatively measure the local interpretability

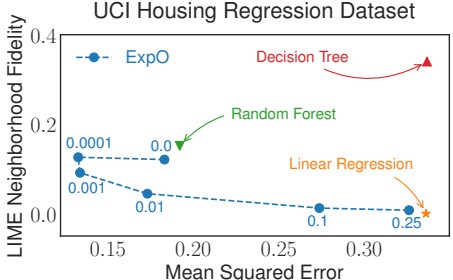 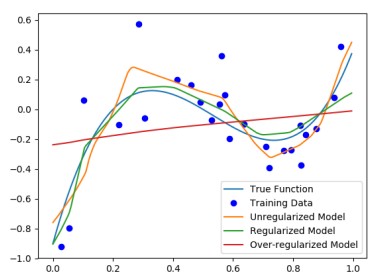

**Figure 1: Left:** Neighborhood Fidelity of LIME-generated explanations (lower is better) vs. predictive error of several models trained on the UCI 'housing' regression dataset. The values in blue denote regularization weight of ExpO; note that it can improve both accuracy and interpretability. **Right:** The effects of ExpO on a model predicting a hypothetical credit rating. The kinks in the unregularized model make local linear explanations have lower fidelity and be less stable to small perturbations. The regularized model is much smoother and more interpretable.

of the model using the standard *fidelity* (Ribeiro et al., 2016; Plumb et al., 2018) and *stability* (Alvarez-Melis and Jaakkola, 2018a) explanation metrics. To make the learned model more amenable to local explanation, ExpO augments the objective function with fidelity- or stability-based regularizers, effectively controlling the degree of local explainability.

The specific contributions of our work are as follows:

1. **Interpretability regularizers.** We introduce two interpretability regularizers associated with the fidelity and stability explanation metrics. The first, ExpO-Fidelity, is designed for semantic features and explainers that directly make predictions, such as those in Ribeiro et al. (2016); Lundberg and Lee (2017); Plumb et al. (2018). The second, ExpO-Stability, is tailored for non-semantic features (*e.g.*, pixels) and explainers that identify features that are influential on a prediction, such as saliency maps (Simonyan et al., 2013). Both regularizers are differentiable and can be used to augment the objective function of an arbitrary model. In Section 3.1, we discuss how they differ from the classical approaches for local approximation and function smoothing.

2. **Generalizable explanation quality.** We analyze the properties of the explanation quality metrics and show that the benefits of our regularization generalize to unseen points. Specifically, we derive a bound on the gap between the fidelity of explanations on training and held out points and connect it with the local variance of the learned model.

**Empirical results.** We evaluate models trained with and without our regularizers on a variety of regression and classification tasks with semantic and image features.[1] Empirically, they slightly improve predictive performance across the nine datasets we consider (seven UCI regression tasks, a medical classification task, and MNIST). From an interpretability perspective, our results demonstrate significant improvement in terms of explanation quality as measured by the fidelity and stability metrics. In particular, our regularizers improved explanation fidelity by at least 25% on the UCI datasets and on the medical classification task while stability on MNIST was improved by orders of magnitude. Additionally, we found, qualitatively, that our regularizers produce models with simpler and more comprehensible explanations. *In summary, black-box explanation systems work better on models trained with ExpO and, as an additional benefit, those models tend to be more accurate.*

## 2 Background and Related Work

Consider a supervised learning problem, where our goal is to estimate a model, $f : \mathcal{X} \mapsto \mathcal{Y}$, $f \in \mathcal{F}$, that maps input feature vectors, $x \in \mathcal{X}$, to targets, $y \in \mathcal{Y}$, and is trained using data, $\{x_i, y_i\}_{i=1}^N$. If the class of functions used for modeling the data is complex, we can

---

[1]The code for our regularizers and all experiments is at: `https://github.com/ForReview11235/CodeForICLR2020`

understand the behavior of $f$ in some neighborhood, $N_x \in \mathcal{P}[\mathcal{X}]$ (where $\mathcal{P}[\mathcal{X}]$ is the space of probability distributions over $\mathcal{X}$), by generating a local *explanation*.

We denote algorithms that produce explanations (*i.e., explainers*) as $e : \mathcal{X} \times \mathcal{F} \mapsto \mathcal{E}$, where $\mathcal{E}$ is the set of possible explanations. The choice of $\mathcal{E}$ generally depends on whether or not $\mathcal{X}$ consists of *semantic features*, and will be defined more precisely next.

## 2.1 SEMANTIC FEATURES

We call features *semantic* if people can reason about them and understand what it means when their values change (*e.g.*, a person's income, the concentration of a chemical, etc.). Consequently, local explanations try to predict how the model's output would change if the input was perturbed (Ribeiro et al., 2016; Lundberg and Lee, 2017; Plumb et al., 2018). Thus, we can define the output space of the explainer as $\mathcal{E}_s := \{g \in \mathcal{G} \mid g : \mathcal{X} \mapsto \mathcal{Y}\}$, where $\mathcal{G}$ is a class of interpretable (typically linear) functions.

**Fidelity metric.** When the explainer's output space is $\mathcal{E}_s$, the explanation is defined as a function $g : \mathcal{X} \mapsto \mathcal{Y}$, and it is natural to evaluate how accurately $g$ models $f$ in a neighborhood $N_x$ (Ribeiro et al., 2016; Plumb et al., 2018):

$$F(f, g, N_x) := \mathbb{E}_{x' \sim N_x}[(g(x') - f(x'))^2], \tag{1}$$

which we refer to as the *neighborhood-fidelity* (NF) metric. This metric is sometimes evaluated with $N_x$ as a point mass on $x$ and we call this version the *point-fidelity* (PF) metric. While Plumb et al. (2018) argued that point-fidelity can be misleading because it does not measure generalization of $e(x, f)$ across $N_x$, it has been used for evaluation in the prior work (Ribeiro et al., 2016; Lundberg and Lee, 2017; Ribeiro et al., 2018) and we report it in our experiments along with the neighborhood-fidelity for completeness.

**Black-box explanation systems.** Various explainers have been proposed to generate local explanations of the form $g : \mathcal{X} \mapsto \mathcal{Y}$, typically assuming that $g$ is linear. In particular, LIME (Ribeiro et al., 2016), one of the most popular black-box explanation systems[2], solves the following optimization problem:

$$e(x, f) := \underset{g \in \mathcal{E}_s}{\arg\min} \, F(f, g, N_x) + \Omega(g), \tag{2}$$

where $\Omega(e)$ stands for an additive regularizer that encourages certain desirable properties of the explanations (*e.g.*, sparsity). LIME's objective function is closely related to the fidelity metric and subsequently to our proposed ExpO-Fidelity regularizer. Consequently, we expect our regularizer to improve the quality of LIME-generated explanations. Our experimental results in Section 4.1 corroborate this hypothesis.

Along with LIME, we consider another black-box explanation tool, called MAPLE (Plumb et al., 2018). It differs substantially from LIME in that its *neighborhood function is learned from the data* rather than specified as a parameter. In our experiments, we evaluate the quality of MAPLE-generated local explanations for models regularized via ExpO-Fidelity, but do not use MAPLE's learned neighborhood function to define ExpO-Fidelity. We view this as a good test case to see how optimizing the fidelity metric for one neighborhood generalizes to another one (see Section 3 for a more detailed discussion of this point). In Section 4.1, we see that regularizing for LIME neighborhoods improves MAPLE's explanation quality.

## 2.2 NON-SEMANTIC FEATURES

Non-semantic features lack an inherent interpretation, with images being a canonical example. When $\mathcal{X}$ consists of non-semantic inputs, we cannot assign meaning to the difference between $x$ and $x'$, hence it does not make sense to explain the difference between the predictions $f(x)$ and $f(x')$. As a result, fidelity is not an appropriate explanation metric. Instead, in this

---

[2]SHAP (Lundberg and Lee, 2017) is another popular method that proposes a theoretically-motivated neighborhood sampling function, but requires explanations to be linear models that act on binary features. This requirement is too limiting in our case.

context, local explanations try to identify which parts of the input are particularly influential on a prediction (Sundararajan et al., 2017). Consequently, we consider explanations of the form $\mathcal{E}_{ns} := \mathbb{R}^d$, where $d$ is the number of features in $\mathcal{X}$.

**Stability metric and saliency maps.** When the explainer's output space is $\mathcal{E}_{ns}$, the explanation is a vector in $\mathbb{R}^d$, and cannot be directly compared to the underlying model itself, as in the case of the fidelity metric. Instead, the focus in this setting is on the degree to which the explanation changes between points in a local neighborhood, which we measure using the *stability metric* (Alvarez-Melis and Jaakkola, 2018a):

$$\mathcal{S}(f, e, N_x) := \mathbb{E}_{x' \sim N_x}[||e(x, f) - e(x', f)||_2^2] \tag{3}$$

Various explainers (Sundararajan et al., 2017; Zeiler and Fergus, 2014; Shrikumar et al., 2016; Smilkov et al., 2017; Adebayo et al., 2018) have been proposed to generate local explanations in $\mathcal{E}_{ns}$, with *saliency maps* (Simonyan et al., 2013) being the approach that we consider in this work. Saliency maps assign importance weights to image pixels based on the magnitude of the gradient of the predicted class with respect to the corresponding pixels.

Recent work on model interpretability emphasizes that more stable explanations tend to be more trustworthy (Alvarez-Melis and Jaakkola, 2018a; Ghorbani et al., 2017; Alvarez-Melis and Jaakkola, 2018b). Note that the stability metric can also be considered in the context of semantic features in addition to the fidelity metric.

## 2.3 RELATED METHODS

A few recently proposed approaches to model interpretability are closely related to our work. First, self-explaining neural networks (SENN) (Alvarez-Melis and Jaakkola, 2018a) (a variation of contextual explanation networks (Al-Shedivat et al., 2017)) is an interpretable by-design approach that additionally (indirectly) optimizes their models to produce stable explanations. Second, "Right For The Right Reasons" (RTFR) (Ross et al., 2017) selectively penalizes gradients of the output with respect to certain input features at some points to discourage their use by the model. Finally, a work concurrent with ours (Lee et al., 2019), which expanded on (Lee et al., 2018), proposed to regularize models of structured data to encourage explainability in a way that is similar to ExpO but differs substantially in how the the target explanation for the regularizer is defined and in how the final objective function is optimized.

From a technical standpoint, SENN and RTFR both assume that the local explanation is close to the first order Taylor approximation of the model at that point. In Section 3.1, we demonstrate how Taylor approximations are often quite different from and more difficult to use than the neighborhood-based local explanations that we use in ExpO. Further, SENN's regularizer requires the neural network to have a very particular structure and, therefore, unlike ExpO, cannot by applied to an arbitrary model. While RTFR's regularization can be used with arbitrary models, it is not directly related to a measure of explanation quality and is defined using specific domain knowledge; on the other hand, ExpO aims to directly improve quality of explanations with respect to a specific metric and does not require domain knowledge. In the Appendix A.1, we compare ExpO to simple $l_1$ and $l_2$ regularization since other baselines are either model specific or require domain knowledge. We found that they do not significantly impact model interpretability.

## 3 EXPLANATION OPTIMIZATION

Running black-box explainers on arbitrary models does not guarantee the quality of the resulting explanations. To address this, we define regularizers that can be added to the loss function and used to train an arbitrary model $f$. Specifically, we want to solve the following optimization problem:

$$\hat{f} := \operatorname*{arg\,min}_{f \in \mathcal{F}} \frac{1}{N} \sum_{i=1}^{N} (\mathcal{L}(f, x_i, y_i) + \gamma \mathcal{R}(f, N_{x_i}^{\mathrm{reg}})) \tag{4}$$

where $\mathcal{L}(f, x_i, y_i)$ is a standard predictive loss (*e.g.*, squared error for regression or cross-entropy for classification), $\mathcal{R}(f, N_{x_i}^{\mathrm{reg}})$ is a regularizer that encourages explainability of $f$ in the neighborhood of $x_i$, and $\gamma > 0$ controls the regularization strength. Because our regularizers are differentiable, we can solve Equation 4 using any standard gradient-based algorithm; in our case, SGD with Adam.

We define $\mathcal{R}(f, N_x^{\mathrm{reg}})$ based on either the neighborhood-fidelity, Eq. (1), or the neighborhood-stability, Eq. (3). In order to compute these metrics exactly, we would need to run an explainer algorithm, $e$; this may be non-differentiable or too computationally expensive to use as a regularizer. Thus, for **ExpO-Fidelity**, we approximate $e$ using a local linear model fit on points sampled from $N_x^{reg}$ (Algorithm 1). For **ExpO-Stability**, we simply require that the model's output not change too much across $N_x^{reg}$ (Algorithm 2).[3]

---

**Algorithm 1** ExpO-Fidelity

**input** $f_\theta$, $x$, $N_x^{\mathrm{reg}}$, $m$
1: Sample: $x'_1, \ldots, x'_m \sim N_x^{\mathrm{reg}}$
2: Compute predictions:
$\qquad \hat{y}_j(\theta) = f_\theta(x'_j)$ for $j = 1, \ldots, m$
3: Produce a linear explanation:
$\qquad \beta_x(\theta) = \arg\min_\beta \sum_{j=1}^{m} (\hat{y}_j(\theta) - \beta^\top x'_j)^2$
**output** $\frac{1}{m} \sum_{j=1}^{m} (\hat{y}_j(\theta) - \beta_x(\theta)^\top x'_j)^2$

---

**Algorithm 2** ExpO-Stability

**input** $f_\theta$, $x$, $N_x^{\mathrm{reg}}$, $m$
1: Sample: $x'_1, \ldots, x'_m \sim N_x^{\mathrm{reg}}$
2: Compute predictions:
$\qquad \hat{y}_j(\theta) = f_\theta(x'_j)$, for $j = 1, \ldots, m$
**output** $\frac{1}{m} \sum_{j=1}^{m} (\hat{y}_j(\theta) - f_\theta(x))^2$

---

**Choosing the neighborhood.** Defining a good regularization neighborhood, requires considering the following. On one hand, we would like $N_x^{\mathrm{reg}}$ to be similar to $N_x$, as used in Eq. 1 or Eq. 3, so that the neighborhoods used for regularization and for evaluation match. On the other hand, we also would like $N_x^{\mathrm{reg}}$ to be consistent with the 'local neighborhood' defined by $e$ internally, which may differ from $N_x$. For LIME, this is not a problem since the internal definition of the 'local neighborhood' is a hyperparameter that we can set. However for MAPLE, the 'local neighborhood' is learned from the data, and hence the regularization and explanation neighborhoods may differ. Ultimately, we left resolving this tension to future work.

**Computational cost.** Algorithm 1 could be prohibitively expensive since the number of samples, $m$, from $N_x^{reg}$, has to be proportional to the dimension of $x$, $d$, resulting in $O(d^3)$ evaluations of $f$ to compute the regularizer at $x$. So we also test a randomized version of Algorithm 1, **ExpO-1D-Fidelity**, that randomly selects one dimension of $x$ to perturb according to $N_x^{reg}$ and penalizes the error of a local linear model along that dimension. This breaks the dependence of the computational cost of the regularizer on $d$ and allows us to compute each gradient step with a $O(1)$ increase in the number of evaluations of $f$.

### 3.1 Understanding the Properties of ExpO

The goal of this section is to compare the behavior of local linear explanations and our regularizers to some existing theoretical function approximations and measures of variance to help develop an intuitive understanding of **ExpO**. First, we compare neighborhood-based local linear explanations to first order Taylor approximations to show that they can have fundamentally very different behaviors. Second, we compare **ExpO-Fidelity** to the Lipchitz Constant (LC) and Total Variation (TV) of the learned function.

**Local explanations vs. Taylor approximations.** A natural question to ask is, *Why should we sample from $N_x$ in order to locally approximate $f$ when there are easier and theoretically motivated approximations?* One possible way to do this is via the Taylor approximation (Alvarez-Melis and Jaakkola, 2018a). The downside of a Taylor approximation-based approach is that such an approximation cannot readily be adjusted to different neighborhood scales and its fidelity and stability strictly depend on the learned function. This can be seen in Figure 2 (left) where the Taylor approximations at two nearby points are both radically different and not faithful to the model outside of an small neighborhood.

**Fidelity regularization and the model's LC or TV.** From a theoretical perspective, our regularizer is similar to controlling the Lipschitz Constant or Total Variation of $f$ across $N_x$ after removing the part of $f$ explained by $e(x, f)$. From an interpretability perspective, there is nothing inherently wrong with having a large LC or TV, which is demonstrated in

---

[3]A similar procedure was explored previously in (Zheng et al., 2016) for adversarial robustness.

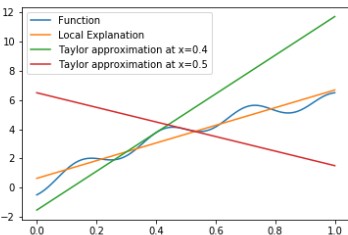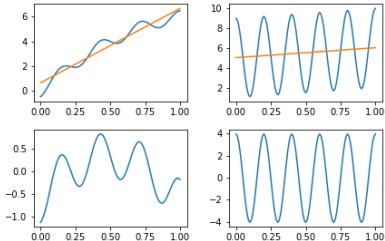

**Figure 2: Left:** A function (blue), its first order Taylor approximations at $x = 0.4$ (green) and $x = 0.5$ (red), and a local explanation of the function (orange) computed with $x = 0.5$ and $N_x = [0, 1]$. **Right (top row):** Two functions (blue) and their local linear explanations (orange). The local explanations were computed with $x = 0.5$ and $N_x = [0, 1]$. **Right (bottom row):** The unexplained portion of the function (residuals).

Figure 2 (right). However, once we take into account what can be explained by $e(x, f)$, then upper bounding any one of ExpO-Fidelity, the LC, or the TV will upper bound the others.

### 3.2 Generalization of Local Linear Explanations

To conclude our analysis, we study the quality of local linear explanations in terms of generalization. Note that ExpO regularization encourages learning models that are explainable in the neighborhoods of each *training point*. However, how would this property generalize to unseen points? We answer this question by providing a generalization bound in terms of neighborhood-fidelity metric for local linear explanations (see Appendix A.2 for derivations).

**Proposition 1** *Let the neighborhood sampling function $N_x$ be characterized by some parameter $\sigma$ (e.g., the effective radius of a neighborhood) and the variance of the trained model $f(x)$ across all such neighborhoods be bounded by some constant $C(\sigma) > 0$. Then, the following bound holds with at least $1 - \delta$ probability:*

$$\mathbb{E}\left[r(f, x)\right] \leq \frac{1}{n} \sum_{i=1}^{n} r(f, x_i) + \sqrt{\frac{C^2(\sigma) \log \frac{1}{\delta}}{2n}} \tag{5}$$

**Remark 2** *The obtained bound tells us that explainable models with smaller local variances across the neighborhoods are likely to have explanations of higher fidelity on the held out points. This further motivates the approximation we used in Algorithm 2.*

## 4 Experimental Results

In our first set of experiments, we demonstrate the effectiveness of ExpO-Fidelity and ExpO-1D-Fidelity on datasets with semantic features using seven regression problems from the UCI collection (Dheeru and Karra Taniskidou, 2017) as well as an in-hospital mortality classification problem. Dataset statistics are in Table 1. Our second experiment demonstrates the effectiveness of ExpO-Stability for creating saliency maps (Simonyan et al., 2013) on MNIST (LeCun, 1998). We found that a model trained with our regularizers is more interpretable than a model trained without them because black-box explainers pro-

| Dataset | # samples | # dims |
|---|---|---|
| autompgs | 392 | 7 |
| communities | 1993 | 102 |
| day | 731 | 14 |
| housing | 506 | 11 |
| music | 1059 | 69 |
| winequality-red | 1599 | 11 |
| MSD | 515345 | 90 |
| SUPPORT2 | 9104 | 51 |
| MNIST | 60000 | 784 |

**Table 1:** Statistics of the datasets.

duce quantitatively better explanations for them; further, they are often more accurate. Finally, we demonstrate qualitatively that the explanations for the regularized model tend to be simpler.

## 4.1 Neighborhood-Fidelity Regularization

We compare models trained with our regularizers to models trained without them. We report accuracy and three interpretability metrics: Point-Fidelity (PF), Neighborhood-Fidelity (NF), and Stability (S). The interpretability metrics are evaluated for two black-box explanation systems: LIME and MAPLE. Consequently, the "MAPLE-PF" label corresponds to the Point-Fidelity Metric for explanations produced by MAPLE.

**Experimental setup.** All of the inputs to the model were standardized to have mean zero and variance one (including the response variable for regression problems). The network architectures and hyper-parameters were chosen using a simple grid search. For the final results, we set $N_x$ to be $\mathcal{N}(x,\sigma)$ with $\sigma = 0.1$ and $N_x^{reg}$ to be $\mathcal{N}(x,\sigma)$ with $\sigma = 0.5$. In the Appendix A.3, we discuss how these values were chosen.

**UCI regression experiments.** The effects of ExpO-Fidelity and ExpO-1D-Fidelity on model accuracy and interpretability are in Table 3. ExpO-Fidelity frequently improved the interpretability metrics by over 50%, with the smallest improvements being around 25%. Further, it lowered the prediction error on the 'communities', 'day', and 'MSD' datasets, which lets us conclude that it has a small positive effect on accuracy as well. ExpO-1D-Fidelity generally had a similar effect on the interpretability metrics.

| Feature | x | Unreg. | ExpO |
|---|---|---|---|
| CRIM | 2.5 | -0.05 | -0.03 |
| INDUS | 1.0 | 0.1 | -0.01 |
| NOX | 0.9 | -0.23 | -0.18 |
| RM | 1.4 | 0.22 | 0.2 |
| AGE | 1.0 | -0.08 | 0.02 |
| DIS | -1.2 | -0.38 | -0.15 |
| RAD | 1.6 | 0.24 | 0.17 |
| TAX | 1.5 | -0.27 | -0.11 |
| PTRATIO | 0.8 | -0.11 | -0.14 |
| B | 0.4 | 0.12 | -0.01 |
| LSTAT | 0.1 | -0.34 | -0.53 |

**Table 2:** LIME's explanation coefficients for unregularized and ExpO-regularized models.

We ran experiments on the 'MSD' dataset[4] to understand the scalability of ExpO to larger tasks. On this dataset, evaluating the interpretability metrics using MAPLE was fairly slow, and hence we only evaluate them using LIME on the first 1000 testing points. Both ExpO-Fidelity and ExpO-1D-Fidelity improved LIME's interpretability metrics by at least 50% and both improved the model accuracy.

**Medical classification experiments.** The 'support2' dataset[5] is used for in-hospital mortality prediction. Because the output layer of our models is the softmax over logits for two classes, we run each explanation system on each of the individual logits. Table 4 presents the results. We observe that ExpO-Fidelity improved the interpretability metrics by 50% or more. ExpO-1D-Fidelity slightly decreased accuracy and improved the interpretability metrics by at least 25%.

**A qualitative example on the UCI 'housing' dataset.** While we have demonstrated quantitatively that black-box explainers produce better explanations for ExpO-regularized models, here we qualitatively analyze the changes in the explanations. More examples, that show similar effects, are available in the Appendix A.4.

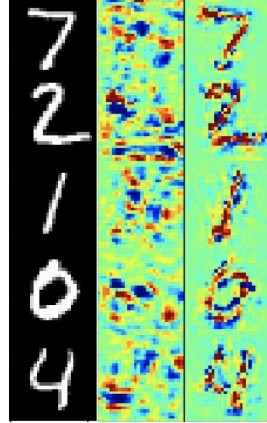

**Figure 3:** Original images (left) and saliency maps of an unregularized (middle) and regularized (right) models.

Consider the UCI 'housing' dataset, a regression problem to predict housing prices in the Boston area. After sampling a random point $x$, we use LIME to generate a local linear explanation at this point for a model trained without regularization ("unregularized explanation") and for a model trained with ExpO-1D-Fidelity ("regularized explanation"). The unregularized and regularized explanations are shown in Table 2.

Quantitatively, using ExpO-1D-Fidelity decreased the LIME-NF metric from 1.15 to 0.02; *i.e.* ExpO produced a model that is much more accurately modeled by the explanation around $x$. Note that the regularized explanation has fewer significant coefficient (those with absolute value greater than 0.1), and hence it is simpler as the effect is attributed to fewer features.

---

[4]The task is to predict release year of song from a set of acoustic features, treated as a regression problem as in Bloniarz et al. (2016)

[5]http://biostat.mc.vanderbilt.edu/wiki/Main/SupportDesc.

**Table 3:** Uregularized model vs. the same model trained with ExpO-Fidelity or ExpO-1D-Fidelity on the UCI regression datasets. Results are shown across 20 trials (with the standard error in parenthesis). Statistically significant improvement ($p = 0.05$) due to Fidelity is denoted in bold and due to 1D-Fidelity is underlined.

| Metric | Regularizer | autompgs | communities | day† ($10^{-3}$) | housing | music | winequality.red | MSD |
|---|---|---|---|---|---|---|---|---|
| **MSE** | None | 0.14 (0.03) | 0.49 (0.05) | 1.000 (0.300) | 0.14 (0.05) | 0.72 (0.09) | 0.65 (0.06) | 0.583 (0.018) |
| | Fidelity | 0.13 (0.02) | **0.46 (0.03)** | **0.002 (0.002)** | 0.15 (0.05) | 0.67 (0.09) | 0.64 (0.06) | **0.557 (0.0162)** |
| | 1D-Fidelity | 0.13 (0.02) | 0.55 (0.04) | 5.800 (8.800) | 0.15 (0.07) | 0.74 (0.07) | 0.66 (0.06) | 0.548 (0.0154) |
| **LIME-PF** | None | 0.040 (0.011) | 0.100 (0.013) | 1.200 (0.370) | 0.14 (0.036) | 0.110 (0.037) | 0.0330 (0.0130) | 0.116 (0.0181) |
| | Fidelity | **0.011 (0.003)** | **0.080 (0.007)** | **0.041 (0.007)** | **0.057 (0.017)** | **0.066 (0.011)** | **0.0025 (0.0006)** | **0.0293 (0.00709)** |
| | 1D-Fidelity | 0.029 (0.007) | 0.079 (0.026) | 0.980 (0.380) | 0.064 (0.017) | 0.080 (0.039) | 0.0029 (0.0011) | 0.057 (0.0079) |
| **LIME-NF** | None | 0.041 (0.012) | 0.110 (0.012) | 1.20 (0.36) | 0.140 (0.037) | 0.112 (0.037) | 0.0330 (0.0140) | 0.117 (0.0178) |
| | Fidelity | **0.011 (0.003)** | **0.079 (0.007)** | **0.04 (0.07)** | **0.057 (0.018)** | **0.066 (0.011)** | **0.0025 (0.0006)** | **0.029 (0.007)** |
| | 1D-Fidelity | 0.029 (0.007) | 0.080 (0.027) | 1.00 (0.39) | 0.064 (0.017) | 0.080 (0.039) | 0.0029 (0.0011) | 0.0575 (0.0079) |
| **LIME-S** | None | 0.0011 (0.0006) | 0.022 (0.003) | 0.150 (0.021) | 0.0047 (0.0012) | 0.0110 (0.0046) | 0.00130 (0.00057) | 0.0368 (0.00759) |
| | Fidelity | **0.0001 (0.0003)** | **0.005 (0.001)** | **0.004 (0.004)** | **0.0012 (0.0002)** | **0.0023 (0.0004)** | **0.00007 (0.00002)** | **0.00171 (0.00034)** |
| | 1D-Fidelity | 0.0008 (0.0003) | 0.018 (0.008) | 0.100 (0.047) | 0.0025 (0.0007) | 0.0084 (0.0052) | 0.00016 (0.00005) | 0.0125 (0.00291) |
| **MAPLE-PF** | None | 0.0160 (0.0088) | 0.16 (0.02) | 1.0000 (0.3000) | 0.057 (0.024) | 0.17 (0.06) | 0.0130 (0.0078) | — |
| | Fidelity | **0.0014 (0.0006)** | **0.13 (0.01)** | **0.0002 (0.0003)** | **0.028 (0.013)** | **0.14 (0.03)** | **0.0027 (0.0010)** | — |
| | 1D-Fidelity | 0.0076 (0.0038) | 0.092 (0.03) | 0.7600 (0.3000) | 0.027 (0.012) | 0.13 (0.05) | 0.0016 (0.0007) | — |
| **MAPLE-NF** | None | 0.0180 (0.0097) | 0.31 (0.04) | 1.2000 (0.3200) | 0.066 (0.024) | 0.18 (0.07) | 0.0130 (0.0079) | — |
| | Fidelity | **0.0015 (0.0006)** | **0.24 (0.05)** | **0.0003 (0.0004)** | **0.033 (0.014)** | **0.14 (0.03)** | **0.0028 (0.0010)** | — |
| | 1D-Fidelity | 0.0084 (0.0040) | 0.16 (0.05) | 0.9400 (0.3600) | 0.032 (0.013) | 0.14 (0.06) | 0.0017 (0.0008) | — |
| **MAPLE-S** | None | 0.0150 (0.0099) | 1.2 (0.2) | 0.0003 (0.0008) | 0.18 (0.14) | 0.08 (0.06) | 0.0043 (0.0020) | — |
| | Fidelity | **0.0017 (0.0005)** | **0.8 (0.4)** | 0.0004 (0.0004) | **0.10 (0.08)** | **0.05 (0.02)** | **0.0009 (0.0004)** | — |
| | 1D-Fidelity | 0.0077 (0.0051) | 0.6 (0.2) | 1.2000 (0.6600) | 0.09 (0.06) | 0.04 (0.02) | 0.0004 (0.0002) | — |

†The relationship between inputs and targets on the 'day' dataset is very close to linear and hence all errors are orders of magnitude smaller than across other datasets.

**Table 4:** Uregularized model vs. the same model trained with ExpO-Fidelity or ExpO-1D-Fidelity on the 'support2' binary classification dataset. Each explanation metric was computed for both the positive and the negative class logits. Results are shown across 10 trials (with the standard error in parenthesis). Improvement due to Fidelity and 1D-Fidelity over unregularized model is statistically significant ($p = 0.05$) for all of the metrics.

| Output | Regularizer | LIME-PF | LIME-NF | LIME-S | MAPLE-PF | MAPLE-NF | MAPLE-S |
|---|---|---|---|---|---|---|---|
| Positive | None | 0.177 (0.063) | 0.182 (0.065) | 0.0255 (0.0084) | 0.024 (0.008) | 0.035 (0.010) | 0.34 (0.06) |
| | Fidelity | **0.050 (0.008)** | **0.051 (0.008)** | **0.0047 (0.0008)** | **0.013 (0.004)** | **0.018 (0.005)** | **0.13 (0.05)** |
| | 1D-Fidelity | 0.082 (0.025) | 0.085 (0.025) | 0.0076 (0.0022) | 0.019 (0.005) | 0.025 (0.005) | 0.16 (0.03) |
| Negative | None | 0.198 (0.078) | 0.205 (0.080) | 0.0289 (0.0121) | 0.028 (0.010) | 0.040 (0.014) | 0.37 (0.18) |
| | Fidelity | **0.050 (0.008)** | **0.051 (0.008)** | **0.0047 (0.0008)** | **0.013 (0.004)** | **0.018 (0.005)** | **0.13 (0.03)** |
| | 1D-Fidelity | 0.081 (0.026) | 0.082 (0.027) | 0.0073 (0.0021) | 0.019 (0.006) | 0.024 (0.007) | 0.16 (0.06) |

**Accuracy (%):** None: $83.0 \pm 0.3$, Fidelity: $83.4 \pm 0.4$, 1D-Fidelity: $82.0 \pm 0.3$.

## 4.2 Stability Regularization

For this experiment, we compared a normally trained convolutional neural network on MNIST to one trained using ExpO-Stability. Then, we evaluated the quality of saliency map explanations for these models. Both $N_x$ and $N_x^{reg}$ where defined as $\mathrm{Unif}(x - 0.05, x + 0.05)$. Both the normally trained model and model traiend with ExpO-Stability achieved the same accuracy of 99%. This demonstrates one of the practical differences between SENN and ExpO: SENN places strict structural constraints on the network and subsequently lowers the testing accuracy to roughly 97%. Quantitatively, training the model with ExpO-Stability decreased the stability metric from 6.94 to 0.0008. Qualitatively, training the model with ExpO-Stability made the resulting saliency maps look much better by focusing them on the presence or absence of certain pen strokes (Figure 3).

## 5 Conclusion

In this work, we have introduced the novel idea of directly regularizing arbitrary models to be more interpretable with respect to a general metric. We contrasted our regularizers to classical approaches for function approximation and smoothing and provided a generalization bound for them. We demonstrated, across a variety of problem settings and explainers, that our regularizers slightly improve model accuracy and improve the interpretability metrics by somewhere from 25% to orders of magnitude. We believe that potential future work may focus on three areas: (1) exploring alternative neighborhood functions, $N_x^{reg}$, that match those used by other black-box explanation systems, (2) exploring how to regularize for non-local interpretability metrics, and finally (3) exploring the interaction between the regularizers and the optimization process, *e.g.*, progressively changing the importance of the regularization during training or using the regularization as an additional training step.

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

## A  APPENDIX

### A.1  COMPARISON OF EXPO TO BASELINE METHODS

Although there are related methods to ExpO, none are necessarily appropriate to act as a baseline for comparison:

- SENN (Alvarez-Melis and Jaakkola, 2018a) requires that the model has a specific structure. ExpO makes no assumptions about the model's structure.
- RTFR (Ross et al., 2017) requires that we have the domain knowledge to specify that "Feature 'i' should not be relevant to the prediction for point 'x' ". ExpO does not require this or any other domain knowledge.
- Finally, the regularizers defined by (Lee et al., 2019) are designed for structured data types. ExpO is designed to work with semantic features or images, so there isn't a common experiment that we could run.

As a result, we consider two standard regularization techniques: $l_2$ and $l_1$ regularization. These regularizers may make the network smoother or simpler (due to sparser weights), which may make it more amenable to local explanation. The results of this experiment are in Table 5; notice that neither of these regularizers had a significant effect on the interpretability metrics compared to ExpO.

| Metric | Regularizer | autompgs | communities | day | housing | music | winequality.red |
|---|---|---|---|---|---|---|---|
| MSE | None | 0.14 | 0.49 | 0.001 | 0.14 | 0.72 | 0.65 |
|  | L2 | 0.13 | 0.47 | 0.00012 | 0.15 | 0.68 | 0.67 |
|  | L1 | 0.12 | 0.46 | 1.7e-05 | 0.15 | 0.68 | 0.67 |
| MAPLE-PF | None | 0.016 | 0.16 | 0.001 | 0.057 | 0.17 | 0.013 |
|  | L2 | 0.015 | 0.17 | 3.2e-05 | 0.05 | 0.17 | 0.02 |
|  | L1 | 0.014 | 0.17 | 1.6e-05 | 0.054 | 0.17 | 0.015 |
| MAPLE-NF | None | 0.018 | 0.31 | 0.0012 | 0.066 | 0.18 | 0.013 |
|  | L2 | 0.016 | 0.32 | 4.3e-05 | 0.058 | 0.17 | 0.021 |
|  | L1 | 0.016 | 0.32 | 2.6e-05 | 0.065 | 0.18 | 0.016 |
| MAPLE-Stability | None | 0.015 | 1.2 | 2.6e-07 | 0.18 | 0.081 | 0.0043 |
|  | L2 | 0.011 | 1.3 | 3.2e-06 | 0.17 | 0.065 | 0.0058 |
|  | L1 | 0.013 | 1.2 | 3e-07 | 0.21 | 0.072 | 0.004 |
| LIME-PF | None | 0.04 | 0.1 | 0.0012 | 0.14 | 0.11 | 0.033 |
|  | L2 | 0.037 | 0.12 | 0.00014 | 0.12 | 0.099 | 0.047 |
|  | L1 | 0.035 | 0.12 | 0.00017 | 0.13 | 0.1 | 0.034 |
| LIME-NF | None | 0.041 | 0.11 | 0.0012 | 0.14 | 0.11 | 0.033 |
|  | L2 | 0.037 | 0.12 | 0.00015 | 0.12 | 0.099 | 0.047 |
|  | L1 | 0.036 | 0.12 | 0.00018 | 0.13 | 0.1 | 0.034 |
| LIME-Stability | None | 0.0011 | 0.022 | 0.00015 | 0.0047 | 0.011 | 0.0013 |
|  | L2 | 0.00097 | 0.032 | 1.7e-05 | 0.004 | 0.011 | 0.0021 |
|  | L1 | 0.0012 | 0.03 | 3e-05 | 0.0048 | 0.011 | 0.0016 |

**Table 5:** Using $l_2$ or $l_1$ regularization has very little impact impact on the interpretability of the learned model.

### A.2  DETAILS ON THE GENERALIZATION OF LOCAL LINEAR EXPLANATIONS

Here, we provide a derivation of the bound (5) on the explanation fidelity. First, we assume that local linear explanations, $\beta_x$, are obtained by solving the ordinary least squares regression problem (as given in Algorithm 1):

$$\beta_x = \left[X'X'^{\top}\right]^{-1} X'f(X'),  \tag{6}$$

where each column of $X'$ denotes a sample from the neighborhood $N_x$ and $f(X')$ is a column-vector of the corresponding function values. The *expected* fidelity of the explanation $\beta_x$ can be computed analytically:

$$r(f,x) = \mathbb{E}_{N_x}\left[f(x')^2\right] - \mathbb{E}_{N_x}\left[f(x')x'\right]^{\top} \mathbb{E}_{N_x}\left[\left[x'x'^{\top}\right]\right]^{-1} \mathbb{E}_{N_x}\left[f(x')x'\right]  \tag{7}$$

where expectation $\mathbb{E}_{N_x}[\cdot]$ is taken with respect to $x'$ over the neighborhood $N_x$. Note the equality in (7) is the expected value of the squared residual between $f(x)$ and the

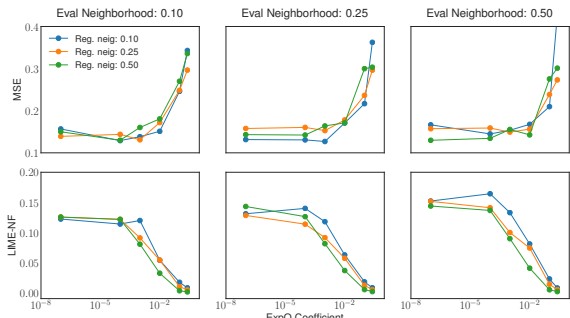

**Figure 4:** A comparison showing the effects of the $\sigma$ parameter of $N_x$ and $N_x^{reg}$ on the UCI Housing dataset. The LIME-NF metric grows slowly with $\sigma$ for $N_x$ as expected. Despite being very large, using $\sigma = 0.5$ for $N_x^{reg}$ is generally best for the LIME-NF metric and possibly for accuracy.

optimal local linear explanation, which is upper-bounded by the variance of the model in the corresponding neighborhood:

$$0 \leq r(f,x) \leq \mathbb{E}_{N_x}\left[f(x')^2\right] - \mathbb{E}_{N_x}\left[f(x')\right]^2 = \mathsf{Var}_{N_x}\left[f(x')\right] \tag{8}$$

For instance, if $f(x)$ is $L$-Lipschitz and the neighborhood $N_x$ is defined a uniform distribution within a $\sigma$-ball centered at $x$, then the variance of $f(x)$ within the neighborhood can be further bounded by $4L^2\sigma^2$, hence $r(f,x) \leq 4L^2\sigma^2$.

For the explanations to generalize, we would like to make sure that the gap between the average fidelity on the training set and the expected fidelity is small with high probability. More formally, the following inequality should hold:

$$\mathbb{P}\left(\mathbb{E}\left[r(f,x)\right] - \frac{1}{n}\sum_{i=1}^{n} r(f,x_i) > \varepsilon\right) < \delta_n(\varepsilon) \tag{9}$$

The following is a restatement of Proposition 1 with a short proof.

**Proposition 3** *Let the neighborhood sampling function $N_x$ be characterized by some parameter $\sigma$ (e.g., the effective radius of a neighborhood) and the variance of the trained model $f(x)$ across all such neighborhoods be bounded by some constant $C(\sigma) > 0$. Then, the following bound holds with at least $1 - \delta$ probability:*

$$\mathbb{E}\left[r(f,x)\right] \leq \frac{1}{n}\sum_{i=1}^{n} r(f,x_i) + \sqrt{\frac{C^2(\sigma)\log\frac{1}{\delta}}{2n}}$$

**Proof.** By assumption, the variance of the model $f(x)$ is bounded in each local neighborhood specified by $N_x$. Then (7) implies that each residual is bounded as $0 \leq r(f,x) \leq C(\sigma)$. Applying Hoeffding's inequality, we get:

$$\mathbb{P}\left(\mathbb{E}\left[r(f,x)\right] - \frac{1}{n}\sum_{i=1}^{n} r(f,x_i) > \varepsilon\right) < \exp\left\{\frac{-2n\varepsilon^2}{C^2(\sigma)}\right\}$$

Inverting the inequality gives us the bound. ∎

### A.3 CHOOSING $\sigma$ FOR $N_x$ AND $N_x^{reg}$

In Figure 4, we see that the choice of $\sigma$ for $N_x$ was not critical (the value of LIME-NF only increased slightly with $\sigma$) and that this choice of $\sigma$ for $N_x^{reg}$ produced slightly more accurate and interpretable models.

## A.4 More Examples of ExpO's Effects

Here, we demonstrate the effects of **ExpO-Fidelity** on more examples from the UCI 'housing' dataset (Table 6). Observe that the same general trends hold true:

- The regularized explanation more accurately reflects the model (LIME-NF metric)
- The regularized explanation generally considers fewer features to be relevant. Again, a feature is 'significant' if its absolute values is 0.1 or greater.
- Neither model appears to be heavily influenced by CRIM or INDUS. The regularized model generally relies more on LSTAT and less on DIS, RAD, and TAX to make its predictions.

| Example Number | Value Shown | CRIM | INDUS | NOX | RM | AGE | DIS | RAD | TAX | PTRATIO | B | LSTAT | LIME-NF |
|---|---|---|---|---|---|---|---|---|---|---|---|---|---|
| 1 | Feature | -0.36 | -0.57 | -0.86 | -1.11 | -0.14 | 0.95 | -0.74 | -1.02 | -0.22 | 0.46 | 0.53 | |
| | Unregularized Explanation | 0.01 | 0.03 | -0.14 | 0.31 | -0.1 | -0.29 | 0.27 | -0.26 | -0.07 | 0.13 | -0.24 | 0.0033 |
| | Regularized Explanation | 0.0 | 0.01 | -0.14 | 0.25 | 0.03 | -0.16 | 0.15 | -0.1 | -0.12 | -0.01 | -0.47 | 0.0033 |
| 2 | Feature | -0.37 | -0.82 | -0.82 | 0.66 | -0.77 | 1.79 | -0.17 | -0.72 | 0.6 | 0.45 | -0.42 | 0.0 |
| | Unregularized Explanation | 0.01 | 0.06 | -0.15 | 0.32 | -0.1 | -0.29 | 0.24 | -0.27 | -0.12 | 0.11 | -0.24 | 0.057 |
| | Regularized Explanation | 0.0 | 0.0 | -0.15 | 0.25 | 0.01 | -0.15 | 0.15 | -0.12 | -0.13 | 0.01 | -0.47 | 0.00076 |
| 3 | Feature | -0.35 | -0.05 | -0.52 | -1.41 | 0.77 | -0.13 | -0.63 | -0.76 | 0.1 | 0.45 | 1.64 | |
| | Unregularized Explanation | -0.01 | 0.06 | -0.16 | 0.29 | -0.08 | -0.31 | 0.27 | -0.27 | -0.11 | 0.1 | -0.18 | 0.076 |
| | Regularized Explanation | -0.03 | -0.01 | -0.13 | 0.19 | -0.0 | -0.15 | 0.14 | -0.11 | -0.12 | 0.0 | -0.43 | 0.058 |
| 4 | Feature | -0.36 | -0.34 | -0.26 | -0.29 | 0.73 | -0.56 | -0.51 | -0.12 | 1.14 | 0.44 | 0.14 | |
| | Unregularized Explanation | 0.02 | 0.06 | -0.18 | 0.29 | -0.1 | -0.34 | 0.31 | -0.21 | -0.09 | 0.12 | -0.27 | 0.10 |
| | Regularized Explanation | -0.02 | 0.01 | -0.13 | 0.21 | 0.02 | -0.16 | 0.17 | -0.11 | -0.12 | -0.0 | -0.47 | 0.013 |
| 5 | Feature | -0.37 | -1.14 | -0.88 | 0.45 | -0.28 | -0.21 | -0.86 | -0.76 | -0.18 | 0.03 | -0.82 | |
| | Unregularized Explanation | 0.02 | 0.08 | -0.17 | 0.33 | -0.11 | -0.36 | 0.29 | -0.27 | -0.08 | 0.1 | -0.28 | 0.099 |
| | Regularized Explanation | -0.0 | -0.0 | -0.14 | 0.26 | 0.0 | -0.16 | 0.15 | -0.11 | -0.15 | 0.01 | -0.47 | 0.0021 |

**Table 6:** More examples of how regularizing a model using **ExpO-Fidelity** affects the explanations. For each example we show, the feature values of the point being explained, the coefficients of the unregularized explanation, and the coefficients of the regularized explanation. Note that the bias terms have been excluded from the explanations. We also report the LIME-NF metric of each explanation.

The same comparison for examples from the UCI 'winequality-red' are in Table 7. We can see that the regularized model depends more on "volatile acidity" and less on "sulphates" while usually agreeing about the effect of "alcohol". Further, it is better explained by those explanations than the unregularized model.

| Example Number | Value Shown | fixed acidity | volatile acidity | citric acid | residual sugar | chlorides | free sulfur dioxide | total sulfur dioxide | density | pH | sulphates | alcohol | LIME-NF |
|---|---|---|---|---|---|---|---|---|---|---|---|---|---|
| 1 | Feature | -0.28 | 1.55 | -1.31 | -0.02 | -0.26 | 3.12 | 1.35 | -0.25 | 0.41 | -0.2 | 0.29 | |
| | Unregularized Explanation | 0.02 | -0.11 | 0.14 | 0.08 | -0.1 | 0.05 | -0.15 | -0.13 | -0.01 | 0.31 | 0.29 | 0.021 |
| | Regularized Explanation | 0.08 | -0.22 | 0.01 | 0.04 | -0.04 | 0.06 | -0.12 | -0.09 | -0.01 | 0.17 | 0.3 | 6.6e-05 |
| 2 | Feature | 1.86 | -1.91 | 1.22 | 0.87 | 0.39 | -1.1 | -0.69 | 1.48 | -0.22 | 1.96 | -0.35 | |
| | Unregularized Explanation | 0.02 | -0.15 | 0.11 | 0.07 | -0.08 | 0.07 | -0.23 | -0.09 | -0.06 | 0.3 | 0.27 | 0.033 |
| | Regularized Explanation | 0.09 | -0.23 | 0.02 | 0.04 | -0.05 | 0.06 | -0.13 | -0.09 | -0.0 | 0.18 | 0.3 | 0.0026 |
| 3 | Feature | -0.63 | -0.82 | 0.56 | 0.11 | -0.39 | 0.72 | -0.11 | -1.59 | 0.16 | 0.42 | 2.21 | |
| | Unregularized Explanation | 0.03 | -0.1 | 0.13 | 0.05 | -0.06 | 0.12 | -0.21 | -0.19 | -0.08 | 0.38 | 0.29 | 0.11 |
| | Regularized Explanation | 0.09 | -0.22 | 0.02 | 0.04 | -0.04 | 0.06 | -0.12 | -0.09 | -0.0 | 0.18 | 0.3 | 8.2e-05 |
| 4 | Feature | -0.51 | -0.66 | -0.15 | -0.53 | -0.43 | 0.24 | 0.04 | -0.56 | 0.35 | -0.2 | -0.07 | |
| | Unregularized Explanation | 0.03 | -0.16 | 0.12 | 0.05 | -0.13 | 0.09 | -0.21 | -0.13 | -0.05 | 0.35 | 0.24 | 0.61 |
| | Regularized Explanation | 0.09 | -0.22 | 0.01 | 0.04 | -0.04 | 0.06 | -0.12 | -0.09 | -0.01 | 0.18 | 0.3 | 6.8e-05 |
| 5 | Feature | -0.28 | 0.43 | 0.1 | -0.65 | 0.61 | -0.62 | -0.51 | 0.36 | -0.35 | 5.6 | -1.26 | |
| | Unregularized Explanation | 0.03 | -0.12 | 0.09 | 0.12 | -0.11 | 0.03 | -0.19 | -0.13 | -0.03 | 0.13 | 0.24 | 0.19 |
| | Regularized Explanation | 0.08 | -0.22 | 0.02 | 0.04 | -0.05 | 0.05 | -0.13 | -0.09 | -0.0 | 0.16 | 0.3 | 0.0082 |

**Table 7:** The same setup, but showing examples for the UCI 'winequality-red' dataset

