# OpenReview forum: "Regularizing Black-box Models for Improved Interpretability"
_ICLR.cc/2020/Conference — Reject_

### Official Review · AnonReviewer1 · 2019-10-22
**Official Blind Review #1**

**Rating:** 6

**Review:**


The paper proposes a new type of regularizer to improve explainability in neural networks. The proposed regularizer is largely based on two metrics, namely fidelity and stability. It optimizes for fidelity and stability as a regularization objective in a differentiable manner.

I would recommend for accept, as the paper shows in its experiments that the explainability of neural networks can be improved with the two proposed regularizer, which outperforms simple baselines of l1 and l2 regularizers. The paper's method is generic, and can be applied to almost all machine learning models with gradient-based optimization, making it helpful to building explainable machine learning systems.

However, I would also like to note that the results in this paper are somewhat unsurprising. The ExpO-Fidelity and ExpO-Stability regularizers can be seen as (almost) directly optimizing the fidelity and stability metric for explainability, so one would naturally expect that models trained with these regularizers will do better on the two metrics above. In addition, I do not see much value in the derivations of Section 3.2. The conclusion that "explainable models with smaller local variances ... are likely to have explanations of higher fidelity" is unsurprising and almost a straightforward claim.

**Experience Assessment:**

I have published one or two papers in this area.

**Review Assessment: Checking Correctness Of Derivations And Theory:**

I assessed the sensibility of the derivations and theory.

**Review Assessment: Checking Correctness Of Experiments:**

I carefully checked the experiments.

**Review Assessment: Thoroughness In Paper Reading:**

I read the paper at least twice and used my best judgement in assessing the paper.

---

> ### Author Response · Authors · 2019-11-11
> **Response to Review #1**
>
> The fidelity and stability metrics have been used to measure interpretability, for local explanations, in previous work.  Although they are proxies for the qualitative concept of “interpretability” in the general sense, they are useful; because a local explanation answers the question “What could I have done differently to get a different outcome?”, a higher fidelity explanation will yield more accurate advice and a more stable explanation will be more trustworthy.  A major contribution of our work is showing that these metrics can in fact be improved using a simple and general method.  Moreover, we disagree with the sentiment that research needs to have a “surprise factor” in order to be impactful.
>
> The purpose of the generalization bound is to demonstrate theoretically that regularizing for fidelity on the training points should improve fidelity on new points.  While not a main contribution of the work, and even though the result itself is not technically difficult to derive, we do think that it is a positive addition to the work.

---

### Official Review · AnonReviewer3 · 2019-10-24
**Official Blind Review #3**

**Rating:** 3

**Review:**

This paper proposes an approach for local post-hoc explanation with introduction of a new regularization that helps regulate the "fidelity" and "stability" of the output explanation (in the style of LIME).  The fidelity regularizer is essentially the squared error of the explainer as compared to the given model in the local neighborhood, whereas the stability regularizer measures the total squared differences between the explanation outcomes between the sample in question and other samples in the local neighborhood.
In the experimental evaluation section, the authors evaluates the performance fo the proposed regularizers, used as part of both LIME and MAPLE, against three interpretability metrics: point fidelity, neighborhood fidelity and stability. The results verify that indeed the use of the regularizers improve the performance of both LIME and MAPLE over the unregularized versions, with respect to the corresponding metrics. This is in a way "expected", since the regularizer used in the method and that in the metric are closely related, and is an unsatisfying aspect of the work.
Using image data, they also demonstrate that qualitatively the use of stability regularizer seems to significantly improve the saliency of the output explanation.
The paper is well written, the proposal is reasonable, but the contribution is modest and experimental evaluation is not entirely convincing.


**Experience Assessment:**

I have read many papers in this area.

**Review Assessment: Checking Correctness Of Derivations And Theory:**

I assessed the sensibility of the derivations and theory.

**Review Assessment: Checking Correctness Of Experiments:**

I assessed the sensibility of the experiments.

**Review Assessment: Thoroughness In Paper Reading:**

I read the paper at least twice and used my best judgement in assessing the paper.

---

> ### Author Response · Authors · 2019-11-11
> **Response to Review #3**
>
> The fidelity and stability metrics have been used to measure interpretability, for local explanations, in previous work.  They are proxies for the qualitative concept of “interpretability” in the general sense, and we would argue are quite useful; because a local explanation answers the question “What could I have done differently to get a different outcome?”, a higher fidelity explanation will yield more accurate advice and a more stable explanation will be more trustworthy.  A major contribution of our work is showing that these metrics can in fact be improved using a simple and general method.  Moreover, we disagree with the sentiment that research needs to have a “surprise factor” in order to be impactful.

---

### Official Review · AnonReviewer2 · 2019-10-25
**Official Blind Review #2**

**Rating:** 1

**Review:**

The paper considers the problem of training black box models for improved interpretability, and  proposes to penalize black-box models at training time using two regularizers that correspond to fidelity and stability explanation metrics. As computing the regularizers exactly is computationally intensive they propose two approximating algorithm. In addition, as the one for fidelity is still prohibitive, a randomized variant is proposed. Connections are established between the regularizers and the model's Lipschitz constant or total variation. A generalization bound is presented for local linear explanations. The proposed approach is evaluated on a variety of datasets.

The paper deals with an important problem and the exposition is clear.  While regularizing deep learning models is a pertinent  direction, I feel the paper makes a couple of overstatements, and overall I am not fully convinced by the approach and empirical evaluation, as outlined below.

- The paper states "recent approaches that claim to overcome this apparent trade-off between prediction accuracy and explanation quality are in fact by-design proposals that impose certain constants on the underlying model families they consider" and that they are addressing this shortcoming. But in fact, the proposed regularizations do exactly the same: they impose certain constraints. Indeed the regularizers encourages models with lower Lipschitz constant or with small total variation across neighborhoods.

- I also find that it is unsurprising that regularizing via fidelity or stability will lead to models with better fidelity/stability so it's an artificial way to yield "improved interpretability" and this is more of an issue because fidelity and stability are kind of proxy metrics to evaluate interpretability.

- It would be important to investigate further the difference between regularization and explanation neighborhoods. This might not be a bad thing which in fact help with generalization.

- Proposition 1 supports algorithm 2, but it is not a given at all that Algorithm 1 will have smaller local variances across neightborhoods and hence might generalize well.  It would be important to proceed with an empirical study of the local variance across neighborhoods for Algorithm 1.

- Computational complexity remains an issue as ExpO-1D-fidelity is performing much worse.

- Comparison against alternative approaches beyond SENN are lacking (e.g. Lee et al 2019, Wang and Rudin,2015 etc).

 Overall I feel that more work is needed to convincingly demonstrate the importante of the proposed approach.


**Experience Assessment:**

I have published in this field for several years.

**Review Assessment: Checking Correctness Of Derivations And Theory:**

I carefully checked the derivations and theory.

**Review Assessment: Checking Correctness Of Experiments:**

I carefully checked the experiments.

**Review Assessment: Thoroughness In Paper Reading:**

I read the paper thoroughly.

---

> ### Author Response · Authors · 2019-11-11
> **Response to Review #2**
>
> This is another by-design approach: "- The paper states ... total variation across neighborhoods."
>
> The distinction being made here is that the other approaches enforce structural restrictions on the model architecture while our regularization is model agnostic.
>
> Further, we believe that you may have misunderstood the connection between our regularization and the Lipschitz constant or the total variation.  The three are only equivalent after the part of the function that can be explained by a local linear explanation has been subtracted; this is discussed when we discuss Figure 2.  As a result, our regularization does not necessarily encourage models with a lower Lipschitz constant or a smaller total variation.
>
>
> Results are Unsurprising: "- I also find ... to evaluate interpretability."
>
> The fidelity and stability metrics have been used to measure interpretability, for local explanations, in previous work.  Although they are proxies for the qualitative concept of “interpretability” in the general sense, they are useful; because a local explanation answers the question “What could I have done differently to get a different outcome?”, a higher fidelity explanation will yield more accurate advice and a more stable explanation will be more trustworthy.  A major contribution of our work is showing that these metrics can in fact be improved using a simple and general method.  Moreover, we disagree with the sentiment that research needs to have a “surprise factor” in order to be impactful.
>
>
> Need to explore different neighborhoods: "- It would be important ... help with generalization. "
>
> We agree that exploring this further is an interesting direction for future work and we said as much.  However, we do not think it is necessary for the current results to be meaningful.
>
>
> What is the effect of ExpO on local variance?:  "- Proposition 1 supports ... for Algorithm 1."
>
> First, we would like to point out that if a regularized model and an ExpO regularized model have the same local variances across neighborhoods, then the fidelity-metric would generalize equally well.  However, the results in the Github repo show that ExpO also reduces this variance.  We excluded this result from the main paper because this variance is neither a measure of accuracy nor interpretability.
>
> Second, the purpose of the generalization bound is to demonstrate theoretically that regularizing for fidelity on the training points should improve fidelity on new points.  The results we provide are on the testing data and consequently are empirical estimates of the true explanation fidelity.  As a result, their importance does not depend on the fact that ExpO also reduced this variance.
>
> Computational Complexity is an issue:  "- Computational complexity ... performing much worse."
>
> We disagree with the characterization that “ExpO-1D-fidelity is performing much worse”.  It still represents a significant improvement over the baseline models.
>
> Comparisons against baselines are missing: "- Comparison against alternative ... Wang and Rudin,2015 etc)."
>
> Lee et al 2019 was a concurrent work with ours and has no publicly available source code.  As discussed in the related work, it differs significantly from our method.
>
> Wang and Rudin, 2015 is a global interpretability method which is fundamentally different from our approach which is related to local interpretability.  These two areas use different metrics and making a direct comparison would be difficult.
>
>
> " Overall I feel that more work is needed to convincingly demonstrate the importante of the proposed approach."
>
> The work demonstrates a simple approach for making post-hoc explanations systems work significantly better.  To the best of our knowledge, there is no other model-agnostic regularization scheme that gives a practitioner the kind of control over local interpretability that we show in Figure 1 Left.

---

> > ### Comment · AnonReviewer2 · 2019-11-13
> > **Additional comments from reviewer#2**
> >
> > -- “The distinction…agnostic. Further, we believe …total variation.”
> >
> > In the present manuscript the authors claim:  "Moreover, recent approaches that claim to overcome this apparent trade-off between prediction accuracy and explanation quality are in fact by-design proposals that impose certain constraints on the underlying model families they consider"
> >
> > The reviewer maintains that this is an overstatement which should be nuanced. Indeed by regularizing, one enforces a constraint, which implicitly imposes some form of restriction on the models being considered. The reviewer did not claim that the proposed regularizers are *equivalent* with LC and TV. As stated by the authors, the proposed regularizers will encourage models that can be represented locally as the sum of a linear explanation plus a term whose LC/TV is controlled. This reflects the general idea that the regularizers constrain the types of models being considered. To reiterate, in the case of local linear explanations, the proposed objective states that the estimator should minimize a predictive loss *subject* to the fact that it should be accurately modeled locally by linear functions in neighborhoods of the training points. Again, this will implicitly constrain the models being produced. The issue is similar for the stability metric.  More generally, there is implicit control, b/c the proposed regularizers introduce coupling between the function being estimated and its explanations, and this coupling ends up impacting the ‘form’ of the estimator itself.
> >
> > --“The fidelity and stability metrics have been used to measure interpretability…Moreover, we disagree with the sentiment that research needs to have a “surprise factor” in order to be impactful.”
> >
> > Surprise factor in-and-of-itself is not the issue. What is an issue is that since regularizing via fidelity or stability will obviously lead to models with better fidelity/stability, it will be crucial to compare against alternative approaches to demonstrate research impact. As stated in a public comment, it should be possible to compare with the work of Lee. Even though Wang and Rudin use a global standpoint, it would be important to benchmark the methods, perhaps even assessing them on both global and local interpretability. Similarly with other related work.
> >
> > Another issue is that the fidelity and stability metrics are proxies; using them in a post-hoc manner to evaluate models might be useful, but it is a more questionable ‘leap’ to *regularize/constrain* the model itself itself using these proxies.
> >
> > --"Computational Complexity is an issue"  The reviewer maintains that this is an issue. ExpO-1D-fidelity is performing much worse than its full-fledged counterpart, and the baseline models considered are insufficient to assess whether ExpO-1D would be valuable at all.

---

### Public Comment · ~Guang-He_Lee1 · 2019-11-08
**Some comments**

Dear authors,

Thank you for this work. We would like to clarify a few things:

1. in Section 2.3 the authors describe their work as "concurrent with" [1,2]. This is a bit strange wording since [1] has been available for more than a year.

2. the authors state that their approach “differs substantially” from [1,2]. The difference seems primarily to be whether the neighborhood is defined as a subset of data space or as a distribution. A subset can be of course defined as samples. So wouldn't the symmetric game formulation in [1, 2] yield the ExpO-Fidelity regularizer if we defined the neighborhood as samples from a distribution together with squared error as deviation and a linear witness class? ExpO-1D-Fidelity also looks like a special case if we restrict the coordinate usage of the linear witness (explainer). Of course restrictions are sometimes valuable but it would be helpful if the authors could clarify these further.


[1] Guang-He Lee, David Alvarez-Melis, and Tommi S. Jaakkola. "Game-Theoretic Interpretability for Temporal Modeling." FAT/ML (ICML workshop), 2018
[2] Guang-He Lee, Wengong Jin, David Alvarez-Melis, and Tommi S. Jaakkola. "Functional Transparency for Structured Data: a Game-Theoretic Approach." ICML 2019.

---

> ### Author Response · Authors · 2019-11-11
> **RE:  Some comments**
>
> In general, we view [1] as addressing a different, but related, problem than our work. The regularization in [1] makes it so that it is easier to explain what the model will predict once it sees the next point in the time series.  This definitely makes the model more interpretable in the general sense (ie, why it’s related).  However, it is conceptually quite different from a local explanation (the focus of ExpO), which explains how the models prediction would change if the time series itself was changed (ie, why it’s different).
>
> We agree that the framework in [2] is defined broadly enough that it includes ExpO as a special case as you describe.  Further, we agree that the applications of [2] to graphs are conceptually similar to local interpretability (unlike the time series experiments in [1]).
>
> However, the optimization in both [1,2] is performed by alternating between updating the witness function and updating the model. This is significantly different from ExpO’s regularizers which are differentiable and require no special optimization strategy.
>
> We claimed to be concurrent with [2] and we noted that [1] preceded our work.  We believe that the claim of concurrency is fair since our work was preprinted before the publication of [2].   We will update the draft to clarify:
> -  That [1] is solving a different, but related, problem and will give more credit to it in the introduction when discussing the idea of regularizing a model to be more interpretable by some metric.
> -  That [2] is solving a similar problem, but with a very different optimization strategy.
>
> Since [2] was concurrent with ExpO and [1] isn’t solving the same problem, we do not think it is necessary to describe ExpO as a special case of either [1,2].

---

### Decision · Program_Chairs · 2019-12-19

**Decision:**

Reject

**Comment:**

This paper investigates a promising direction on the important topic of interpretability; the reviewers find a variety of issues with the work, and I urge the authors to refine and extend their investigations.